# Chronic Calcifying Pancreatitis Associated with Secondary Diabetes Mellitus and Hepatosplenic Abscesses in a Young Male Patient: A Case Report

Cristina Maria Marginean [1], Mihaela Popescu [2,†], Corina Maria Vasile [3,*] , Mihaela Stanciu [4], Iulian Alin Popescu [1], Viorel Biciusca [1] , Daniela Ciobanu [1,*], Amelia Dobrescu [5], Larisa Daniela Sandulescu [6] , Simona Bondari [7], Marian Sorin Popescu [8] and Paul Mitrut [1]

[1] Department of Internal Medicine, University of Medicine and Pharmacy of Craiova, 200349 Craiova, Romania
[2] Department of Endocrinology, University of Medicine and Pharmacy of Craiova, 200349 Craiova, Romania
[3] Department of Pediatric Cardiology, "Marie Curie" Emergency Children's Hospital, 041451 Bucharest, Romania
[4] Department of Endocrinology, Faculty of Medicine, Lucian Blaga University of Sibiu, 550169 Sibiu, Romania
[5] Department of Genetics, University of Medicine and Pharmacy of Craiova, 200349 Craiova, Romania
[6] Research Center of Gastroenterology and Hepatology, University of Medicine and Pharmacy, 200349 Craiova, Romania
[7] Department of Radiology, University of Medicine and Pharmacy of Craiova, 200349 Craiova, Romania
[8] Ph.D. School Department, University of Medicine and Pharmacy of Craiova, 200349 Craiova, Romania
* Correspondence: corina.vasile93@gmail.com (C.M.V.); elada192@yahoo.com (D.C.)
† These authors contributed equally to this work.

**Abstract:** Background: Chronic pancreatitis (CP) has been described as a multifactorial, ongoing inflammatory condition of the pancreas of varying intensity that produces persistent pain, leading to exocrine and endocrine insufficiency and a decreased lifespan. Currently, there are three primary forms of chronic pancreatitis: chronic autoimmune pancreatitis (steroid-sensitive pancreatitis), chronic obstructive pancreatitis, and chronic calcific pancreatitis, the latter being closely related to excessive alcohol consumption for one or even two decades before the onset of symptoms. Case report: We present the case of a 29 year old man who required medical attention for a significant unintentional weight loss and a history of upper abdominal pain. Blood tests revealed substantial abnormalities, and the patient was admitted for further investigation. CT and MRI confirmed the presence of a pancreatic pseudocyst and extensive pancreatic parenchymal calcifications and revealed multiple hepatosplenic microabscesses of fungal etiology. Conclusions: Chronic calcifying pancreatitis is a complex clinical entity that can lead to secondary diabetes due to progressive destruction of the pancreatic parenchyma. Protein malnutrition, caused by malabsorption syndrome, immune cell dysfunction, and a high glucose environment caused by diabetes mellitus, may create a state of immunodeficiency, predisposing the patient to opportunistic infections.

**Keywords:** chronic calcifying pancreatitis; computed tomography; hepatosplenic abscesses; diabetes mellitus

## 1. Introduction

Chronic pancreatitis (CP) has been described as a multifactorial, ongoing inflammatory condition of the pancreas of varying intensity that produces persistent pain, subsequently leading to exocrine and endocrine insufficiency and a decreased lifespan [1–4]. Possible long-term complications include diabetes and pancreatic cancer. The incidence of chronic pancreatitis in European countries varies between 5 and 10 per 100,000 inhabitants, with a median survival of 20 years [5].

Currently, there are three primary forms of chronic pancreatitis: chronic autoimmune pancreatitis (steroid-sensitive pancreatitis), chronic obstructive pancreatitis, and chronic

calcific pancreatitis, the latter being closely related to excessive alcohol consumption for one or even two decades before the onset of symptoms [5–7]. Chronic calcific pancreatitis is a rare cause of secondary diabetes. According to one study, 118 (20.1%) out of 587 patients with chronic pancreatitis developed diabetes. The same study points out risk factors for endocrine failure in this group of patients: old age, smoking, parenchymal and pancreatic duct calcifications, and pancreatic duct strictures [8].

Mycotic abscesses are most often caused by Candida spp. Hepatic and splenic fungal abscesses can appear in immunocompromised subjects and patients suffering from malignancies. Neutropenia is the most common feature in individuals with high susceptibility to developing abscesses of fungal etiology. Fungemia is a direct predictor of mortality, and prevention of its onset by adequate antifungal treatment is life-saving in most cases [9,10].

## 2. Case Report

We report the case of a 29 year old patient who presented with involuntary weight loss (approximately 10 kg in the months before examination) associated with polydipsia, polyuria, pale stools, and frequent diarrheal episodes in the past six months. The patient complained of severe upper abdominal pain for the past two years. In addition, our patient had a positive history of moderate to intense daily alcohol consumption. There had been no reported cases of pancreatic disease in close relatives.

Upon clinical examination, a discrete general pallor and dry skin were observed. The patient was underweight (BMI = 17.6 kg/m$^2$). Blood pressure and heart rate were normal. The abdomen was soft to the touch without rigidity. Upper abdominal pain was present, both spontaneous and on palpation. Physical examination revealed a palpable mass, immobile on respiration, with a smooth surface and firm consistency.

Rapid blood tests in the emergency department revealed an increased blood glucose level of 281.59 mg/dL. The patient underwent hospitalization and further additional laboratory investigations (Table 1). Serum amylase and lipase were both within normal limits. No ketone bodies were present in the urine. Examination of stool revealed a high-fat content and a low fecal elastase level.

**Table 1.** Laboratory test results of the patient.

| Parameter | Admission | During Hospitalization | Discharge |
|---|---|---|---|
| Leukocyte count ($\times 10^3/\mu$L) | 2.60 | 4.54 | 6.86 |
| Neutrophil count ($\times 10^3/\mu$L) | 1.50 | 2.84 | 3.86 |
| Lymphocyte count ($\times 10^3/\mu$L) | 0.80 | 1.37 | 2.42 |
| Platalet count ($\times 10^3/\mu$L) | 198.00 | 148.80 | 200.20 |
| Erythrocyte count ($\times 10^6/\mu$L) | 2.88 | 3.26 | 3.07 |
| Hemoglobin (g/dL) | 10.00 | 10.79 | 10.51 |
| Chloride (mmol/L) | 91.00 | 98.00 | 98.00 |
| Potassium (mmol/L) | 2.80 | 4.00 | 3.50 |
| Sodium (mmol/L) | 129.00 | 134.00 | 136.00 |
| Calcium (mmol/L) | - | 6.80 | - |
| Glucose (mg/dL) | 281.59 | 126.00 | 137.42 |
| Total serum protein (g/dL) | 5.60 | 5.60 | 6.00 |
| Serum albumin (g/dL) | 2.4 | 2.8 | 2.8 |
| ALT (U/L) | 63.30 | 424.00 | 166.00 |
| AST (U/L) | 23.20 | 472.00 | 35.00 |
| GGT (U/L) | 314.00 | 452.00 | 381.00 |
| BUN (mg/dL) | 35.00 | 20.00 | - |
| Creatinine (mg/dL) | 0.54 | 0.57 | - |
| CRP (mg/dL) | - | 24.00 | - |
| ESR (mm/h) | - | 23.00 | - |
| Amylase (U/L) | 44.00 | 56.00 | - |
| C-peptide (ng/mL) | - | 0.23 | - |
| PTH (pg/mL) | - | 28.46 | - |

ALT, alanine aminotransferase; AST, aspartate aminotransferase; GGT, gamma-glutamyl transferase; BUN, blood urea nitrogen; CRP, C-reactive protein; ESR, erythrocyte sedimentation rate; PTH, parathormone.

Abdominal ultrasonography showed extensive pancreatic calcifications and pseudo-cysts with a diameter of 3 cm (Figure 1).

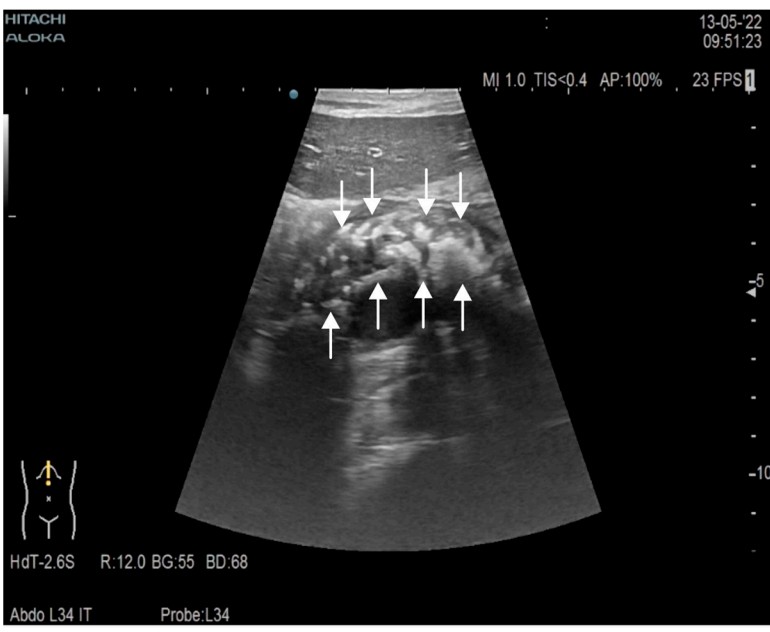

**Figure 1.** Pancreatic calcifications and pseudocyst revealed on US scan. White arrows: extensive pancreatic calcifications and pseudocysts.

We performed a contrast CT scan, which detected a bilateral pleural effusion (maximum width 1.1 cm) and a pericardial effusion (full width 2.4 cm). Multiple round, hypodense, hypointense, and infracentimetric lesions were described in the liver. Comparable features were also found in the spleen. The lesions were interpreted as hepatosplenic microabscesses.

There were widespread calcifications throughout the pancreatic parenchyma and a pseudocyst measuring 3.3 by 3.2 cm (Figure 2). Wirsung duct dilatation was also present. These findings were interpreted as chronic calcifying pancreatitis.

A contrast-enhanced MRI scan also confirmed all the findings as mentioned above and, in addition, suggested a mycotic etiology of hepatosplenic abscesses, describing multiple hepatosplenic lesions with T2 hyper signal, restricted diffusion on DWI sequences and correlating decreased ADC values (Figure 3).

Clinical, biological, and imaging evidence strongly supported the diagnosis of chronic calcifying pancreatitis. However, the exact etiology was difficult to establish.

Primary hyperparathyroidism, a known condition associated with pancreatic calcification, could be excluded as both parathyroid hormone (PTH) and serum calcium values were within normal limits. In our patient's case, a diagnosis of cystic fibrosis could be safely ruled out because there was no confirmed history of recurrent respiratory infections in childhood, imaging investigations were negative for bronchiectasis, and a sweat test showed normal chloride levels.

Good insulin response, absence of ketone bodies, and a serum C-peptide of 0.23 ng/mL pleaded for diabetes secondary to pancreatic injury.

Imaging findings and a positive result of antifungal treatment suggested a fungal etiology.

Fluconazole was administered for abscesses for 14 days. Clinical improvement was slow but favorable, with subsequent contrast ultrasonography confirming complete remission (Figure 4).

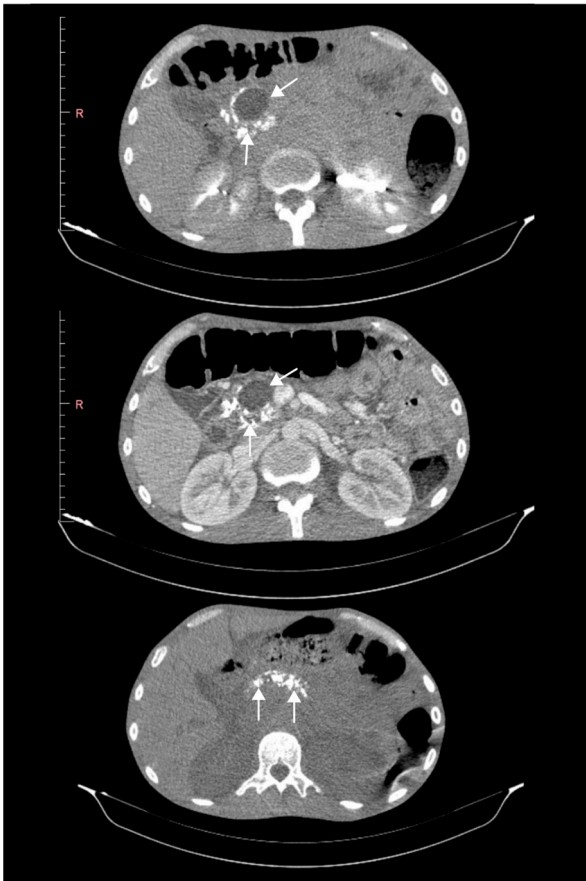

**Figure 2.** Pancreatic calcifications and pseudocyst revealed on CT scan. White arrows: widespread calcifications throughout the pancreatic parenchyma and a pseudocyst.

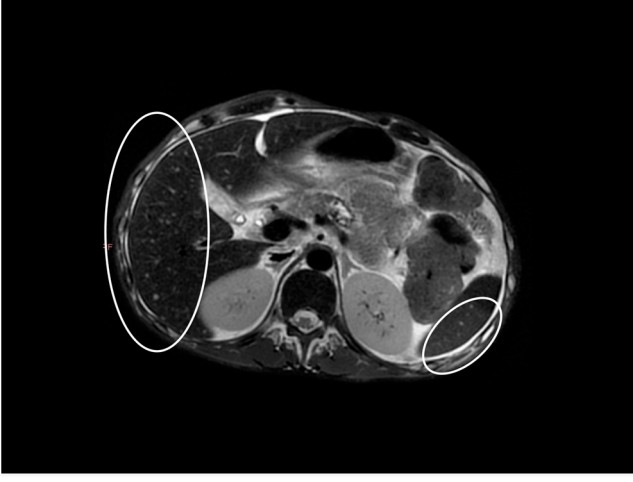

**Figure 3.** Multiple T2 hyper signal hepatosplenic lesions on MRI. White circles: multiple hepatosplenic lesions.

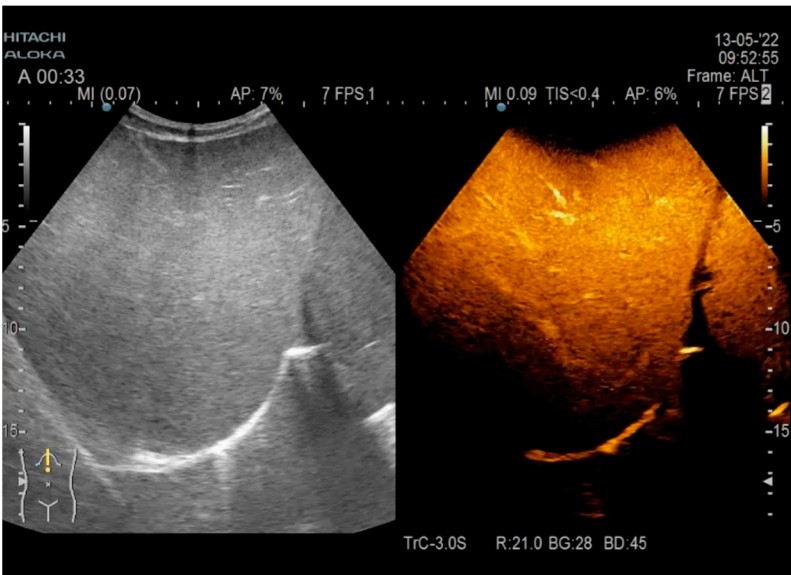

**Figure 4.** Total remission after fluconazole treatment on ultrasound findings.

Regular human insulin was used to control blood glucose values, with desirable results. The patient underwent fluid and pancreatic enzyme replacement therapy with a favorable outcome.

## 3. Discussion

Pancreatic calcification is mainly caused by chronic calcific pancreatitis [9], while obstructive and autoimmune pancreatitis rarely causes calcification [5].

Since chronic alcoholic calcific pancreatitis is defined by a consumption threshold for alcohol that is rather arbitrary, a significant fraction of idiopathic cases is incorrectly labeled as alcoholic or vice versa [7]. Furthermore, most gastroenterologists tend to assume alcoholic etiology whenever there is a confession of ethanol consumption.

In general, the onset of the first symptoms of chronic alcoholic pancreatitis appears in the fourth and fifth decades (with a median age of 44 years). In contrast, early-onset chronic idiopathic pancreatitis symptoms start during adolescence (with a median age of 19 years) [7].

At the time of diagnosis, several patients with chronic alcoholic pancreatitis have calcifications and exocrine and endocrine pancreatic dysfunction. Such features are rarely present at the onset of idiopathic chronic pancreatitis [7].

Although the relationship between pancreatitis and hyperparathyroidism remains controversial, recent studies suggest it is essential to investigate PTH and blood calcium in patients with pancreatitis [10].

Fungal infections are often associated with immunosuppressed patients, especially those with neutropenia. On admission, our patient had a neutrophil number of $1.5 \times 10^3/\mu$L. Therefore, he was highly susceptible to infections with fungal microorganisms. This immune impairment is explained by protein malnutrition, secondary to malabsorption syndrome, and diabetes mellitus.

The less common etiological factors of pancreatic calcifications include cystic fibrosis, neuroendocrine tumors, pancreatic adenocarcinoma, intraductal papillary mucinous neoplasia, pancreatoblastoma, and calcified pancreatic metastases [9].

Mucoviscidosis is caused by defects in the cystic fibrosis transmembrane conductance regulator (CFTR) gene, leading to the production of an aberrant CFTR protein and chloride channel abnormalities [11]. These channels can be found in the apical membrane's cutaneous, reproductive, respiratory, and digestive epithelia. This condition has an autosomal recessive inheritance [11]. Calcification of the pancreas has been observed in cystic fibrosis patients, usually distended pancreatic ducts [12]. Some of these patients, especially those

with severe pancreatic dysfunction, have also shown atrophy of the pancreas and diffuse calcification of the pancreatic parenchyma [8]. Infracentimetric pancreatic cysts have also been described, but pancreatic cysts (pancreas entirely replaced by cysts) are considered rare [9].

Pancreatic neuroendocrine tumors (PNETs) originate from the endocrine pancreas and represent less than 5% of pancreatic neoplasms. Multiple endocrine neoplasia type I, tuberous sclerosis complex, neurofibromatosis type 1, and von Hippel–Lindau syndrome are familial disorders that include the presence of PNETs. However, more than 90% of islet cell tumors occur sporadically [13]. The current studies found in the literature divide PNETs into two groups: hyperfunctioning and non-hyperfunctioning islet cell tumors. Insulin-producing and non-insulin-producing types are two categories that may further divide hyper-functional tumors [14]. Pancreatic calcifications have been observed most in non-hyper functional PNETs [14]; one study found calcifications in 6 out of 25 tumors (22%) [15]. Calcification caused by PNETs has a specific morphology that may be easily differentiated from chronic pancreatitis—discrete, nodular, and coarse calcifications localized in one anatomical part of the pancreas [16].

Calcification in primary gastrointestinal carcinoma is well known. However, in the literature, there are rare reports of its occurrence in pancreatic adenocarcinoma, most commonly found in pancreatic neoplasms [17]. In a study that analyzed CT scans of 100 patients with pancreatic calcifications [18], only four were diagnosed with adenocarcinoma. Moreover, in three of them, the neoplastic process was superimposed on pre-existing chronic pancreatitis. In these patients, calcifications were found within the neoplastic parenchyma, suggesting that these were features of chronic pancreatitis, not adenocarcinoma per se.

Intraductal papillary mucinous neoplasia (IPMN) is a tumor that causes cystic dilatation of the central pancreatic duct, the secondary ducts, or a combination of the two. Branch-duct IPMNs are generally benign, while main- and mixed-duct IPMNs are usually malignant. The most common presentation of a patient with IPMN is mild to moderate acute pancreatitis [19]. In 20% of IPMNs, calcifications are visible on CT scans, with a higher frequency in larger lesions [20]. Pancreatic calcifications detected in patients with IPMN appear as unstructured deposits within the pancreatic ducts. Their formation has been explained by the presence of mucin and its high calcium salt content [21].

Pancreatoblastoma is a rare primary neoplasm of the pancreas, mainly affecting children between 1 and 8 years of age [22]. The pancreas head is the typical site of tumor development. Histopathological analysis of the tissue reveals embryonal pancreatic tissue [23]. Abdominal pain, vomiting, constipation, and early satiety are among the complaints of patients suffering from pancreatoblastoma. The tumor may secrete an adrenocorticotrophic hormone, thus associating the syndrome of inadequate antidiuretic hormone secretion and Cushing's syndrome [22]. Imagistically, the tumor appears as a heterogeneous solid mass with a clearly defined outline, showing calcification in 30% of cases [24]. The calcification pattern shows calcifications clustered within the tumor and along the margins [9].

Pancreatic metastases are not distributed to a specific part of the pancreas and usually originate from lung (22.7%) or renal (30.3%) carcinoma [25]. Calcified metastases of the pancreas have rarely been reported and have had a renal, colonic, or ovarian origin. Calcifications have not been related to lung metastases [9,26].

In their review, Javadi et al. outlined a broad spectrum of pathological conditions that can affect the pancreas due to calcification. A pattern recognition approach involving the morphological features of the pancreatic lesion and the pattern of calcification may be helpful to make an appropriate differential diagnosis and achieve a specific diagnosis, as the management of these lesions varies. Diagnostic imaging studies and relevant laboratory findings should be interpreted in the clinical context. Occasionally, a definitive diagnosis cannot be made by imaging alone, and tissue sampling should be performed, as several of these features may have overlapping characteristics [10].

In their recent study, Bilgin et al. compared MRI and MRCP findings of the pancreas in diabetic patients with or without pancreatic exocrine insufficiency, which was determined

by low fecal elastase levels. Their study demonstrated that most (79%) patients with diabetes combined with PEI had more than three abnormal side branches [27].

The recent study by Shetty R et al., revealed that the mean age at diagnosis of patients with chronic pancreatitis was 24.2 years, ranging from 18 to 37 years; the mean BMI of these patients was 21.1 kg/m$^2$. On average, the duration of the disease was 5.6 years. Almost all patients (98.6%) complained of pain; 25.7% of patients had diabetes, and 41.1% had steatosis. The mean fecal elastase level was 82.5–120 µg elastase/g, ranging from 5 to 501 µg elastase/g. A total of 50 patients (71.4%) had a low FE 1 level, and the rest had an average FE 1 level. This study also showed no association between patient age, gender, BMI, and age of disease onset and FE 1 levels. Those with low FE 1 had a significantly longer total disease duration and a longer duration of diabetes than those with average FE 1 [28].

## 4. Conclusions

Chronic calcific pancreatitis is a complex clinical entity that can lead to secondary diabetes due to progressive destruction of the pancreatic parenchyma. Protein malnutrition, caused by malabsorption syndrome, immune cell dysfunction, and a glucose-rich environment caused by diabetes mellitus, can develop a state of immunodeficiency, predisposing the patient to opportunistic infections.

A challenge in the presented case was the patient's high susceptibility to opportunistic infections, translated as hepatosplenic microabscesses. Malnutrition and diabetes mellitus created a suitable immunological terrain for fungal proliferation. The imaging team could have easily overlooked the fungal infection, as its clinical features were entirely masked by the base pathology, underlining the importance of thoroughly investigating vulnerable patients.

**Author Contributions:** Conceptualization, C.M.M., M.P. and P.M.; methodology, C.M.M., C.M.V. and I.A.P.; software, M.S.P.; validation, C.M.M., M.P., P.M., M.S. and C.M.V.; formal analysis, C.M.M.; investigation, I.A.P., L.D.S. and D.C.; resources, S.B., V.B. and A.D.; data curation, M.S., D.C. and M.S.P.; writing—original draft preparation, C.M.M., M.P. and P.M.; writing—review and editing, C.M.V.; visualization, C.M.V. and M.S.; supervision, D.C. and P.M.; project administration, C.M.M. All authors have read and agreed to the published version of the manuscript.

**Funding:** This research received no external funding.

**Institutional Review Board Statement:** The study was conducted in accordance with the Declaration of Helsinki and approved by the Institutional Review Board (or Ethics Committee) of the University of Medicine and Pharmacy of Craiova (approval no. 158/09.08.2022).

**Informed Consent Statement:** Informed consent was obtained from all subjects involved in the study. Written informed consent has been obtained from the patient to publish this paper.

**Conflicts of Interest:** The authors declare no conflict of interest.

## Abbreviations

| | |
|---|---|
| CP | chronic pancreatitis |
| PTH | parathyroid hormone |
| DWI | diffusion-weighted MRI |
| ADC | apparent diffusion coefficient MRI |
| CFTR | cystic fibrosis transmembrane conductance regulator gene |
| PNETs | pancreatic neuroendocrine tumors |
| IPMN | intraductal papillary mucinous neoplasia |
| CT | computerized tomography |
| MRI | magnetic resonance imaging |
| MRCP | magnetic resonance cholangiopancreatography |
| PEI | pancreatic exocrine insufficiency |
| FE1 | fecal elastase 1 |
| BMI | body mass index |

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
