# Peer review of "Chronic Calcifying Pancreatitis Associated with Secondary Diabetes Mellitus and Hepatosplenic Abscesses in a Young Male Patient: A Case Report"

_gastroent, doi:10.3390/gastroent13030031_

Round 1

Reviewer 1 Report

I reviewed the case report titled "Chronic calcifying pancreatitis associated with secondary diabetes mellitus and hepatosplenic abscesses in a young male patient".

In my opinion, extensive changes have to be made to make the report publishable. 

ABSTRACT: No references are needed in the abstract. 

INTRO: should be shorter, just 1-2 paragraphs and to the point

CASE REPORT: check CARE guidelines for structuring the report https://www.equator-network.org/reporting-guidelines/care/

Author Response

Dear Reviewer,

Thank you very much for your review and constructive suggestions.

R1:

I reviewed the case report titled "Chronic calcifying pancreatitis associated with secondary diabetes mellitus and hepatosplenic abscesses in a young male patient".

In my opinion, extensive changes have to be made to make the report publishable.

Q1:ABSTRACT: No references are needed in the abstract.

A1: We apologize for our negligence. We have changed it accordingly. Thank you for pointing this out.

Q2:INTRO: should be shorter, just 1-2 paragraphs and to the point

A2: Thank you for the suggestion. Given the complexity of our case, we found it difficult to limit ourselves to a brief introduction. At your suggestion, we have revised it accordingly.

Q3:CASE REPORT: check CARE guidelines for structuring the report https://www.equator-network.org/reporting-guidelines/care.

A3: Thank you for your advice. We have modified according to CARE guidelines.

Reviewer 2 Report

The authors presented a case with chronic calcific pancreatitis complicated by diabetes mellitus and hepatosplenic abscesses of fungal etiology. Notable labs include decreased neutrophil count that gradually improved over the hospital course. Patient was treated with fluconazole with improvement in clinical status. This is an interesting case.

1.            Introduction should also include discussion of other complications of chronic pancreatitis, particularly hepatic and splenic abscesses with which the patient in this case report presented.

2.            Line 125-129 should be moved to Discussion.

3.            Line 130-131, why did regular human insulin fail to achieve good control of blood glucose?

4.            The Discussion focuses on the etiology of pancreatic calcification, but it should also further discuss the unique/unusual aspects of the presentation and management of this case. What are the important clinical pearls and teaching points from this case? 

Author Response

Dear Reviewer,

Thank you very much for your review and constructive suggestions.

R2:

The authors presented a case with chronic calcific pancreatitis complicated by diabetes mellitus and hepatosplenic abscesses of fungal etiology. Notable labs include decreased neutrophil count that gradually improved over the hospital course. Patient was treated with fluconazole with improvement in clinical status. This is an interesting case.

Q1:Introduction should also include discussion of other complications of chronic pancreatitis, particularly hepatic and splenic abscesses with which the patient in this case report presented.

A1: We have added in the introduction part this data at your suggestion. Thank you.

Mycotic abscesses are most often caused by Candida spp. Hepatic and splenic fungal abscesses can appear in immunocompromised subjects and patients suffering from malignancies. The most common feature identified in individuals with high susceptibility for developing abscesses of fungal etiology is neutropenia.  Fungemia is a direct predictor of mortality and prevention of its onset by adequate antifungal treatment is life-saving in most cases [5,6].

Q2:Line 125-129 should be moved to Discussion.

A2: Thank you for the advice. We have made the appropriate changes.

Q3:  Line 130-131, why did regular human insulin fail to achieve good control of blood glucose?

A3: We apologise for our negligence. It was a redaction error, human insulin was able to regulate blood glucose levels in our patient.

Regular human insulin has been used to control blood glucose values, with desirable results. The patient underwent fluid therapy and pancreatic enzyme replacement therapy, with a favorable outcome.

Q4 :The Discussion focuses on the etiology of pancreatic calcification, but it should also further discuss the unique/unusual aspects of the presentation and management of this case. What are the important clinical pearls and teaching points from this case?

A4: Thank you for suggesting this. We have added the most important aspects of our case.

A particular challenge encountered in the presented case was the patient’s high susceptibility to opportunistic infections, translated as hepato-splenic microabscesses. Malnutrition and diabetes mellitus created the right immunological terrain for fungal proliferation. The fungal infection could have been easily overlooked if missed by the imaging team, as its clinical features were completely masked by the base pathology, underlining the great importance of a thorough investigation of vulnerable patients.

Round 2

Reviewer 1 Report

There are some minor errors that have to be corrected. 

Table 1: Lukocyte count --> Leukocyte count; You should revise the use of commas and dots to the English language. Thrombocytes -->Platelet count

Figure 1: legends say that it is a CT image, while it is US

Line 121: Clinical improvement (not progression) was slow

Line 203: 30.3; % is missing

Line 216: PEI should be spelled out

Line 218: what is IEP?

Line 223: FE should be used at first mentioning fecal elastase, then the abbreviation should be used. 

Reviewer 2 Report

The authors have made changes and improved their manuscript. I have some minor suggestions:

1. Please proofread the manuscript and make sure the typos are corrected. 

2. Fig 1-4 can be enhanced by including arrows pointing to the relevant findings. 
